# Combined Radiomodifying Effect of Fucoidan from the Brown Alga *Saccharina cichorioides* and Pacificusoside D from the Starfish *Solaster pacificus* in the Model of 3D Melanoma Cells

**DOI:** 10.3390/biom13030419

**Published:** 2023-02-23

**Authors:** Olesya S. Malyarenko, Timofey V. Malyarenko, Roza V. Usoltseva, Alla A. Kicha, Natalia V. Ivanchina, Svetlana P. Ermakova

**Affiliations:** G.B. Elyakov Pacific Institute of Bioorganic Chemistry, Far Eastern Branch of the Russian Academy of Sciences, 159 100-let Vladivostok Ave., 690022 Vladivostok, Russia

**Keywords:** brown alga, *Saccharina cichorioides*, fucoidan, starfish, *Solaster pacificus*, pacificusoside D, SK-MEL-2 cells, 3D cell culture, radiomodifying effect, apoptosis, DNA degradation

## Abstract

Cancer is one of the main causes of human mortality worldwide. Despite the advances in the diagnostics, surgery, radiotherapy, and chemotherapy, the search for more effective treatment regimens and drug combinations are relevant. This work aimed to assess the radiomodifying effect and molecular mechanism of action of fucoidan from the brown alga *Saccharina cichorioides* (ScF) and product of its autohydrolysis (ScF_AH) in combination with pacificusoside D from the starfish *Solaster pacificus* (SpD) on the model of viability and invasion of three-dimension (3D) human melanoma cells SK-MEL-2. The cytotoxicity of ScF (IC_50_ JB6 Cl41 > 800 µg/mL; IC_50_ SK-MEL-2 = 685.7 µg/mL), ScF_AH (IC_50_ JB6 Cl41/SK-MEL-2 > 800 µg/mL), SpD (IC_50_ JB6 Cl41 = 22 µM; IC_50_ SK-MEL-2 = 5.5 µM), and X-ray (ID_50_ JB6 Cl41 = 11.7 Gy; ID_50_ SK-MEL-2 = 6.7 Gy) was determined using MTS assay. The efficiency of two-component treatment of 3D SK-MEL-2 cells was revealed for ScF in combination with SpD or X-ray but not for the combination of fucoidan derivative ScF_AH with SpD or X-ray. The pre-treatment of spheroids with ScF, followed by cell irradiation with X-ray and treatment with SpD (three-component treatment) at low non-toxic concentrations, led to significant inhibition of the spheroids’ viability and invasion and appeared to be the most effective therapeutic scheme for SK-MEL-2 cells. The molecular mechanism of radiomodifying effect of ScF with SpD was associated with the activation of the initiator and effector caspases, which in turn caused the DNA degradation in SK-MEL-2 cells as determined by the Western blotting and DNA comet assays. Thus, the combination of fucoidan from brown algae and triterpene glycoside from starfish with radiotherapy might contribute to the development of highly effective method for melanoma therapy.

## 1. Introduction

Melanoma is a highly aggressive form of skin cancer with an increasing prevalence worldwide [1]. In 2020, an estimated 325,000 new cases of melanoma were diagnosed all over the world and 57,000 people died from the disease [2]. Even with recent advancements in melanoma therapies—namely, immunotherapies such as ipilimumab, targeted therapies such as vemurafenib or combination therapies such as polychemotherapy, polyimmunotherapy, and biochemotherapy—the management of advanced melanoma is very challenging [3,4].

Surgery, radiation, and chemotherapy are conventional therapeutic modalities for the treatment of the patients with malignant neoplasms [5]. Unfortunately, none of them can satisfy the requirements of clinical practice and does not fully meet the variety of tasks that make up the program for the radical treatment of cancer. The most significant problems of chemotherapy include the development of the complications, due to the toxicity of anticancer drugs, as well as the formation of drug resistance [6].

Radiation therapy is also one of the leading methods of cancer treatment, allowing to reduce the size of the tumor or to completely destroy it. According to the recommendations of the World Health Organization, radiation therapy is indicated for more than 70% of cancer patients in independent radical plan or as a component of combined and complex therapy. Despite the advances in the radiation therapy of cancer, the problem of high-dose radiation toxicity and radioresistance of cancer cells continues to be one of the most important for oncology [7].

That is why the optimization of cancer therapy and the search for more effective treatment regimens and drug combinations are challenged. The prospects for the combination of radiation and chemotherapy (radiochemotherapy) of cancer are selectively influencing the sensitivity of malignant tumors to chemo- and radiotherapeutic effects and expanding the boundaries of the therapeutic window. In this regard, the search and investigations of compounds as anticancer drugs, radioprotectors or radiosensitizers, their dosage, and the optimal scheme of their administration are relevant.

The sulfated polysaccharides of the brown algae, fucoidans, their derivatives and glycosylated compounds of starfish were proved to have potent cytotoxic, anti-proliferative, anti-migratory, and anti-metastatic activities against panel of human cancer cells that make them perspective candidates as anticancer drugs, radioprotectors or radiosensitizers to improve cancer therapy [8,9,10,11,12].

Fucoidan from the brown alga *Saccharina cichorioides* was reported to be almost pure fucan, containing the main chain of 1,3-linked α-L-fucopyranose residues with a small degree of 1,4-linked α-L-fucopyranose residues. A small amount of single α-L-fucose residues were in the branches at the position 2. Sulfate groups occupied position 2 and 4 of fucopyranose residues [13]. Fucoidan from *S. cichorioides* was shown to effectively inhibit TPA-induced neoplastic transformation of the mouse epidermal cells JB6 Cl41 [14], the proliferation and colony formation of human colorectal carcinoma cells HT-29 and DLD-1, melanoma RPMI-7951, and breast cancer T-47D cell lines [15,16] and enhanced the anti-proliferative activity of resveratrol in the human colorectal carcinoma cell line HCT 116 [17]. The molecular mechanism of its anticancer action was associated with the regulation of expression and activity of mitogen activated protein kinases (MAPK) and the induction of apoptosis.

Starfish, being by nature active predators, serve as a rich source of various low-molecular weights biologically active metabolites: peptides, sterols, polar steroids, and their glycosides, carotenoids, anthraquinone pigments, as well as sphingolipids and their derivatives [18,19,20,21]. Occasionally, there are also some other classes of low-molecular weight compounds, for example, triterpene glycosides, alkaloids or some toxins that are apparently ingested in starfish with food. Recently, new triterpene glycoside, pacificusoside D, was isolated from the Far Eastern starfish *Solaster pacificus* [22]. Pacificusoside D was reported to exhibit strong cytotoxic and significant colony-forming inhibitory activities against the human melanoma cells SK-MEL-2, SK-MEL-28, and RPMI-7951 [22].

In the past, in vitro cytotoxic activity of synthetic and natural compounds was studied using established cancer cell lines grown as two-dimensional (2D) cultures characterized by a rapid, uncontrolled growth phenotype. However, 2D cell cultures are not capable of mimicking the complexity and heterogeneity of clinical tumors as in vivo tumors grow in a three-dimensional (3D) conformation with a specific organization and architecture that a 2D monolayer cell culture cannot reproduce. Three-dimensional (3D) growth of immortalized established cell lines or primary cell cultures possess several in vivo features of tumors such as cell-cell interaction, hypoxia, drug penetration, response and resistance, and production/deposition of extracellular matrix [23].

We reported herein the results of the study on the radiomodifying effect of fucoidan from the brown algae *S. cichorioides* and its derivative obtained by autohydrolysis in combination with triterpene glycoside, pacificusoside D, from the starfish *S. pacificus* on the model of viability and invasion of three-dimensional (3D) human melanoma cells SK-MEL-2. Moreover, our research was revealed the molecular mechanism of combined radiomodifying effect of investigated biomolecules, which is associated with the induction of apoptosis and DNA fragmentation in SK-MEL-2 cells.

## 2. Materials and Methods

### 2.1. Reagents

Organic solvents, inorganic acids and salts, sodium hydroxide and trifluoroacetic acid (TFA) were commercial products (Laverna-Lab, Moscow, Russia). Standards (mannose, rhamnose, glucose, galactose, xylose, and dextrans) were purchased from Sigma-Aldrich (St. Louis, MO, USA). Macro-Prep DEAE was obtained from Bio Rad Laboratories, Inc. (Hercules, CA, USA), and Amberlite CG-120—from Serva Electrophoresis GmbH (Heidelberg, Germany).

Phosphate buffered saline (PBS), dimethyl sulfoxide (DMSO), L-glutamine, penicillin-streptomycin solution (10,000 U/mL, 10 µg/mL), Minimum Essential Medium Eagle (MEM), gelatin from porcine skin type A were purchased from “Sigma-Aldrich” company (St. Louis, MO, USA). MTS reagent—3-[4,5-dimethylthiazol-2-yl]-2,5-diphenyltetrazolium bromide was purchased from “Promega” (Madison, WI, USA). Trypsin, fetal bovine serum (FBS), agarose, and protein marker “Page ruler plus prestained protein ladder” were purchased from “Thermo Fisher Scientific” (Waltham, MA, USA).

### 2.2. Fucoidan from the Brown Alga S. cichorioides and Its Derivative Obtained by Autohydrolysis

#### 2.2.1. The Brown Alga

The sample of the brown alga *Saccharina cichorioides* (Sc) was collected in August 2021 from Peter the Great Bay, Sea of Japan (Russia). Fresh algal biomass was powdered and pretreated with 70% aqueous ethanol (1:10 *w*/*v*) at room temperature for 10 days. Defatted alga was air-dried.

#### 2.2.2. Isolation of Fucoidan from the Brown Alga *S. cichorioides*

The fucoidan ScF was isolated from the brown alga (100 g) by the earlier described method [16] with a yield of 4 g.

#### 2.2.3. Autohydrolysis of Fucoidan

Aliquot (20 mL) of the solution of fucoidan ScF (5 mg/mL) was converted to the H^+^-form using an Amberlite CG-120 column (200–400 mesh, H^+^-form, 1 × 5 cm) and left at 37 °C for 72 h. The mixture was neutralized with 5% NH_4_OH solution, concentrated on a rotary evaporator to 2.5 mL, and fractionated by EtOH (1:4). The supernatant was discarded, and the pellet ScF_AH was solubilized in water and lyophilized, with a yield of 56 mg.

#### 2.2.4. The Determination of Structural Characteristics of Fucoidan and Its Derivative

Total carbohydrates were quantified by the phenol-sulfuric acid method [24].

The monosaccharide composition after polysaccharide hydrolysis using 2 M TFA (6 h, 100 °C) and obtaining of alditol acetate derivatives was determined by gas-liquid chromatography (GLC).

The degree of sulfation was determined by using the BaCl_2_—gelatin method [25].

Molecular weight was determined by size-exclusion chromatography (SEC), using an Agilent 1100 Series HPLC instrument (“Agilent Technologies”, Waldbronn, Germany) equipped with a refractive index detector and series-connected SEC columns, Shodex OHpak SB-805 HQ and OHpak SB-803 HQ, (“Showa Denko”, Tokyo, Japan). Elution was performed with 0.15 M NaCl aqueous solution at 40 °C, with a flow rate of 0.4 mL/min. The dextrans of 5, 10, 50, 80, 250, 410, and 670 kDa (“Sigma-Aldrich”, St. Louis, MO, USA) were used as reference standards.

Nuclear magnetic resonance (NMR) spectra were obtained using an Avance DPX-500 NMR spectrometer (Bruker BioSpin Corporation, Billerica, MA, USA) at 35 °C. The concentration of the samples was 10 mg of polysaccharide/600 µL of D_2_O.

### 2.3. Triterpene Glycoside from the Starfish S. pacificus

#### 2.3.1. Starfish

Specimens of *S. pacificus* Djakonov, 1938 (order Valvatida, family Solasteridae) were collected at a depth of 10–20 m in the Sea of Okhotsk near Iturup Island during the research vessel *Akademik Oparin’s* 42nd scientific cruise in August 2012. Species identification was carried out by B.B. Grebnev (G.B. Elyakov Pacific Institute of Bioorganic Chemistry of the FEB RAS, Vladivostok, Russia). A voucher specimen (no. 042-112) is on deposit at the marine specimen collection of the G.B. Elyakov Pacific Institute of Bioorganic Chemistry of the FEB RAS, Vladivostok, Russia.

#### 2.3.2. Isolation of Pacificusoside D from the Starfish *S. pacificus*

Pacificusoside D was obtained from the starfish *S. pacificus* by methods published earlier and was pure according to NMR, MS, TLC, and HPLC data [22].

### 2.4. Biological Activity Study

#### 2.4.1. The Preparation of Investigated Compounds

The vehicle control is the spheroids treated with equivalent volume of PBS or DMSO (final concentration was less than 0.5%) for all presented experiments.

Fucoidan from *S. cichorioides* ScF (ScF) or its derivative ScF_AH (AH) were dissolved in sterile PBS, filtered by 0.22 µm membrane (“Millipore”, Billerica, MA, USA) to prepare stock concentrations of 20 mg/mL.

Pacificusoside D from *S. pacificus* (SpD) was dissolved in sterile DMSO to prepare stock concentrations of 20 mM.

#### 2.4.2. Cell Culture Conditions

The mouse epidermal cells JB6 Cl41 (ATCC^®^ CRL-2010™) were obtained from the American Type Culture Collection (Manassas, VA, USA). Human melanoma cells SK-MEL-2 (ATCC^®^ HTB-68™) were provided by the Shared research facility “Vertebrate cell culture collection” (Saint-Petersburg, Russia). JB6 Cl41 and SK-MEL-2 cells were cultured in MEM medium supplemented by 5% and 10% heat-inactivated FBS, respectively, and 1% penicillin-streptomycin solution at 37 °C in humidified atmosphere containing 5% CO_2_. The passage number was carefully controlled, and the mycoplasma contamination was monitored on a regular basis.

#### 2.4.3. Formation of Spheroids (3D Cell Culture)

Spheroids were formed by liquid overlay technique as described earlier [26]. JB6 Cl41 and SK-MEL-2 cells (5.0 × 10^3^) were inoculated in 1.5% agarose layer and cultured in 200 µL of complete MEM/5% FBS or MEM/10% FBS medium, respectively, for 72 h at 37 °C in a 5% CO_2_ incubator. Spheroid integrity, diameter, and volume were analyzed with the aid of ZOE ™ Fluorescent Cell Imager (“Bio Rad”, Hercules, CA, USA) and ImageJ software bundled with 64-bit Java 1.8.0_112 (“NIH”, Bethesda, MD, USA).

#### 2.4.4. X-ray Exposure of Spheroids

After of JB6 Cl41 and SK-MEL-2 spheroids were formed, they were exposed to X-ray irradiation using “XPERT 80 X-ray” system (KUB Technologies, Inc, Milford, CT, USA). The absorbed dose of radiation was measured by DRK-1 X-ray radiation clinical dosimeter (Axelbant LLK, Moscow, Russia).

#### 2.4.5. Viability of Spheroids

##### Individual Cytotoxic Effect of Fucoidan, Its Derivative, Pacificusoside D, and X-ray

The individual effect of ScF, ScF_AH, SpD, or X-ray on viability of 3D cell culture was determined by MTS method. Briefly, formed spheroids of JB6 Cl41 and SK-MEK-2 cells were treated by replacing 100 µL of supernatant with complete medium containing ScF at 100–800 µg/mL, ScF_AH at 100–800 µg/mL, SpD at 0.5–100 µM, or X-ray at 2–16 Gy for 24 h. Then 15 µL of MTS reagent was added to the each well with spheroids and incubated for 3 h at 37 °C in 5% CO_2_. The absorbance of each well was measured at 490/630 nm using Power Wave XS microplate reader (“BioTek”, Wynusky, VT, USA).

The concentration of compounds (IC_50_) or dose of X-ray radiation (ID_50_) at which they exert half of its maximal inhibitory effect on cell viability was calculated by AAT-Bioquest^®^ online calculator [27].

The selective index (SI) of investigated compounds was calculated by the following equation: SI = IC_50_ against normal cells JB6 Cl41 / IC_50_ against cancer cells SK-MEL-2. Therapeutic index of (TI) of X-ray = ID_50_ against normal cells JB6 Cl41 / ID_50_ against cancer cells SK-MEL-2 [28].

##### Combined Cytotoxic Effect of Investigated Compounds and X-ray

Three-dimensional JB6 Cl41 and SK-MEL-2 cells were treated by compounds or X-ray in fixed non-toxic concentration/dose ratios, which were chosen according to their IC_50_ and ID_50_, and the sequence of spheroids treatment with the investigated compounds and X-ray. Namely, the combined treatment by ScF (50, 100, and 200 µg/mL) together with SpD (1 µM) or X-ray (2 Gy); ScF_AH (50, 100, and 200 µg/mL) together with SpD (1 µM) or X-ray (2 Gy); SpD (1 µM) together with ScF (50, 100, and 200 µg/mL) or ScF_AH (50, 100, and 200 µg/mL) or X-ray (2 Gy); and X-ray (2 Gy) together with ScF (50, 100, and 200 µg/mL) or ScF_AH (50, 100, and 200 µg/mL) or SpD (1 µM) was realized. The metabolic activity of JB6 Cl41 and SK-MEL-2 spheroids after the combined treatment was measured by the MTS assay, as described above in Section “Individual cytotoxic effect of fucoidan, its derivative, pacificusoside D, and X-ray”.

#### 2.4.6. Three-Dimensional Invasion of Spheroids

##### Preparation of Gelatin Coating Plates

Some (0.5 mL) gelatin (0.1% (*v*/*v*)) was loaded onto each well of 6-well plates and left for stacking at room temperature for 2 h. Then, the residual unbound amount of gelatin was aspirated, and the wells were washed twice with PBS. The gelatin layer was blocked by 0.5 mL of BSA (1% (*w*/*v*)) at room temperature for 1 h. After blocking, the residual BSA was aspirated, and 1 mL of culture medium supplemented with 2% FBS was added to each well.

##### Three-Dimensional Spheroid-Based Invasion

After SK-MEL-2 spheroids were formed, they were treated by replacing 100 µL of supernatant with complete medium containing individual compounds ScF (50, 100, and 200 µg/mL) or X-ray (2 Gy) or SpD (1 µM) for 72 h and their combination (FXS): pre-treatment of spheroids by ScF (50, 100, and 200 µg/mL) for 24 h, then exposure by X-ray (2 Gy) for 24 h, and finally, treatment with SpD (1 µM) for 24 h. Spheroid from each treatment was transferred into gelatin coating plates (3 replicates per condition) using a multichannel pipette in a 100 µL of appropriate culture medium. The spheroids were maintained at 37 °C, in a 5% CO_2_ incubator for 72 h. Photo of 3D spheroids (40 × 500 μm scale) were made at time points of 0 and 72 h with the aid of a microscope, Motic AE 20. Quantitative assessment of the invasion of spheroids was carried out using the ImageJ program and was determined as the difference between total area invaded by cells leaving the spheroid and the area of the spheroid as described previously [29].

#### 2.4.7. Western Blotting

Formed SK-MEL-2 spheroids were treated by individual compounds and their combination FXS as described above in the Section “Spheroid-based invasion”. The spheroids (*n* = 6) from each treatment were collected to the 1.5 mL microcentrifuge tubes and left for sedimentation for 10 min. The spheroids were washed by PBS and then lysed by 1X lysis buffer (“Cell Signaling Technology”, Danvers, MA, USA) according to the manufacturer’s protocol. Proteins were extracted, separated by 10% or 12% polyacrylamide gel electrophoresis and blotted (20–40 µg). The membranes were treated with the primary antibodies and secondary antibodies from rabbit or mouse (“Sigma-Aldrich”, St. Louis, MO, USA), according to the manufacturer’s protocol. Protein bands were visualized using an enhanced chemiluminescence reagent (ECL) (“Bio-Rad”, Hercules, CA, USA), according to the manufacturer’s protocol. Relative band density was measured using Image Lab™ Software 4.1.

#### 2.4.8. DNA Comet

Formed SK-MEL-2 spheroids were treated by individual compounds ScF or X-ray or SpD and their combination FXS as described above in the Section “3D Spheroid-based invasion”. The spheroids (*n* = 3) from each treatment were collected in 1.5 mL Eppendorf microcentrifuge tubes and left for sedimentation for 10 min. Then, the supernatants were removed and the spheroids were disassociated by 100 µL of 0.25% trypsin/0.5 mM EDTA for 5 min. The spheroids were thoroughly resuspended in complete culture media and the number of cells was counted by TC10™ Automated Cell Counter (“Bio Rad”, Hercules, CA, USA). The cells of disassociated spheroids (2 × 10^3^/mL) were mixed with 1 mL of 1% low-melting-point agarose and 1 mL of cell suspension was applied to agarose-precoated slides. Cell lysis and agarose gel electrophoresis were performed under alkaline conditions at 15 V for 30 min at 4 °C, as described previously [30]. The agarose gel was stained with ethidium bromide (25 µg/mL) for 30 min, and DNA comets were visualized using a ZOE™ Fluorescent Cell Imager fluorescent microscope.

#### 2.4.9. Combination Index (CI) Calculation

The dose-effects of ScF, ScF_AH, SpD, X-ray and their combinations were calculated by “Compusyn software” (ComboSyn, Inc., Paramus, NJ, USA) using the median effect method described by Chou [31].

All assays were performed in at least three independent experiments. Results are expressed as the mean ± standard deviation (SD). Statistical procedures were performed using one-way ANOVA and Tukey’s HSD tests with * *p* < 0.05, ** *p* < 0.01, and *** *p* < 0.001.

## 3. Results

### 3.1. Fucodan from the Brown Alga S. cichorioides and Its Modification by Autohydrolysis

The sulfated polysaccharide, fucoidan, ScF was isolated from the Far Eastern brown alga *S. cichorioides* according to an individual isolation scheme, including the extraction of water-soluble polysaccharides, hydrophobic chromatography on Polychrome-1, and anion exchange chromatography on DEAE-macroprep, as described in our previous work [16]. Structural characteristics of the obtained fucoidan (monosaccharide composition; molecular weight (Mw); content of sulfate and acetate groups; impurities of proteins; polyphenols) were determined using classical chemical methods, HPLC, and NMR spectroscopy. ScF was confirmed to be a highly sulfated (35%) α-L-fucan with Mw of 554 kDa.

According to NMR spectroscopy, fucoidan ScF was confirmed to consist mainly of (1→3)-linked α-L-fucopyranose residues and a small amount of (1→4)-linked fucopyranose residues with branches at position 2 in the form of single α-L-fucose. Sulfate groups are found in positions 2 and 4 (Appendix A).

In the present work, in order to investigate a polysaccharide with a regular structure and reveal the impact of structural characteristics on its activity, a fucoidan derivative ScF_AH was obtained using the autohydrolysis method (mild acid hydrolysis with the participation of polysaccharide’s sulfate groups). It was determined that the Mw and degree of sulfation of fucoidan’s derivative ScF_AH were decreased until 251 kDa and 20%, respectively, compared to native fucoidan ScF with Mw of 554 kDa and degree of sulfation of 35%. Based to ^13^C NMR data ScF_AH was found to be a 4-sulfated α-(1→3)-L-fucan (Appendix A).

### 3.2. Pacificusoside D from the Starfish S. pacificus

The structure of pacificusoside D from the starfish *S. pacificus* was previously investigated and published [22]. This triterpene glycoside contains a holostane type of aglycon, having a 16β-OAc and 7(8)-double bond in the nucleus and *trans-*Δ^22,24^ side chain, and carbohydrate chain, including five monosaccharides (two xyloses, quinovose, glucose, and 3-*O*-methyl-glucose) attached to C-3 of the aglycon. All these sugars are of D-series and attached to each other and aglycon by β-glycosidic bonds (Figure 1).

### 3.3. The Effect of Fucoidan, Its Derivative, Pacificusoside D, and X-ray on the Viability of SK-MEL-2 Spheroids

Firstly, the concentration of fucoidan from the brown alga *S. cichorioides* (ScF), its derivative (ScF_AH), pacificusoside D (SpD), or dose of X-ray irradiation (X-ray), required for the inhibition of cell viability by 50% (IC_50_ or ID_50_), was determined by MTS assay on the model of 3D cultures (spheroids) of mouse epidermal JB6 Cl41 cells and human melanoma SK-MEL-2 cells. It was calculated that IC_50_ of ScF; SpD; and ID_50_ X-ray was 685.7 µg/mL; 5.5 μM; and 11.7 Gy, respectively, after 24 h of treatment of SK-MEL-2 spheroids (Figure 2a,c,d). The product of autohydrolysis of fucoidan ScF_AH did not exert cytotoxic effect under the same experimental conditions (Figure 2b). Normal mouse epidermal cells JB6 Cl41 were used to calculate the selectivity index (SI) of pacificusoside D (SpD) or therapeutic index (TI) of X-ray, which indicate the safety and efficacy of used treatment modalities. The SI of SpD was 4 and TI of X-ray—1.74, respectively. SI for fucoidan and its derivative was not calculated, because they had no cytotoxic activity against JB6 Cl41 spheroids at concentrations up to 800 µg/mL.

### 3.4. The Combined Effect of Fucoidan or Its Derivative with Pacificusoside D and X-ray on the Viability of SK-MEL-2 Spheroids

To study the combined effects of investigated compounds, 3D SK-MEL-2 cells were treated by ScF (50, 100, and 200 µg/mL), ScF_AH (50, 100, and 200 µg/mL), SpD (1 µM), and X-ray (2 Gy) in fixed low non-toxic concentration at which they alone had a slight effect on the cell viability of spheroids of normal epidermal JB6 Cl41 cells and melanoma SK-MEL-2 cells (the percentage of inhibition of cell viability was less than 20%) (Figure 3, Appendix A). Additionally, the combined effect of investigated compounds and X-ray on viability of SK-MEL-2 spheroids was determined in dependence of the sequences of treatment by investigated compounds, as well as the time of incubation of the spheroids.

First of all, the effectiveness of two-component combination FS, FX, AHS, AHX, SF, SAH, SX, XF, XAH, and XS was checked on the model of the viability of SK-MEL-2 spheroids (Figure 3 and Figure 4).

It was shown that the pre-treatment of SK-MEL-2 spheroids by fucoidan ScF at concentrations of 50, 100, and 200 µg/mL in combination with pacificusoside D SpD (FS combination) caused the cell viability suppression by 40%, 45%, and 47%, respectively, compared to the PBS-treated spheroids (control) (Figure 3a), while the pre-treatment by ScF in combination with X-ray irradiation (FX combination)—by 38%, 45%, and 55%, respectively, compared to control (Figure 3b). The type of two-component combined effect of FS (CI = 0.34; 0.35; 0.16) and FX (CI = 0.67; 0.6; 0.51) was found to be synergistic (Figure 3c).

The product of autohydrolysis of fucoidan ScF_AH in combination with SpD (AHS combination) or ScF_AH with X-ray (AHX combination) slightly affected the viability of 3D SK-MEL-2 cells (Figure 3d,e). In this case, antagonism or additivity of the combined effect was observed (Figure 3f).

The combined treatment of SK-MEL-2 spheroids by pacificusoside D SpD with fucoidan ScF (SF combination) (Figure 4a,d) or with X-ray (SX combination) (Figure 4c,d), as well as X-ray with ScF (XF combination) (Figure 4e,h) or with SpD (XS combination) (Figure 4g,h) led to the inhibition of the cell viability, but to a lesser extent than FS and FX combinations. The effectiveness of SpD in combination with ScF_AH (SAH combination) (Figure 4b,d) and X-ray with ScF_AH (XAH combination) (Figure 4f,h) was not detected at the same concentrations and time point of treatment.

Based on the results of two-component combined treatment of SK-MEL-2 spheroids, three-component combinations (FXS, SXF, XFS, and XSF) of fucoidan from *S. cichorioides* ScF (F) with pacificusoside D from *S. pacificus* SpD (S) and X-ray (X) were selected and studied their effectiveness (Figure 5).

Fucoidan ScF at 50, 100, and 200 µg/mL was found to sensitize 3D SK-MEL-2 cells to X-ray irradiation (2 Gy) and enhance the inhibitory effect of pacificusoside D SpD (1 µM), that led to a significant decrease in the viability of SK-MEL-2 spheroids by 55%, 63%, and 68%, respectively, compared to control (Figure 5a). The combination index (CI) or fraction affected (Fa) indicated the degree of drug interaction was 0.44, 0.35, and 0.33 or 0.45, 0.37, and 0.33, respectively, which confirm the synergism of the combined radiomodifying effect of ScF and SpD on viability of SK-MEL-2 spheroids (Figure 5c). It should be noted that combined radiomodifying effect of other tested combinations (SXF, XFS, and XSF) was less pronounced (Figure 5b,d,e). The type of interactions was nearly additive as determined by the Chou–Talalay method for drug combination (Figure 5f).

Thus, in this study, the most effective and safety scheme of combined treatment causing significant inhibition of the viability of SK-MEL-2 spheroids was found to be the pre-treatment of SK-MEL-2 spheroids by fucoidan ScF, then their exposure with X-ray and consequent inhibition of 3D cell viability by pacificusoside D SpD (FXS combination). Additionally, it was confirmed that the viability of normal epidermal cells JB6 Cl41 was not suppressed significantly under their treatment by the FXS combination (Appendix A). That is why the FXS combination for treatment SK-MEL-2 spheroids was chosen for further investigation of the combined radiomodifying effect of investigated compound on 3D invasion of melanoma cells.

### 3.5. The Combined Radiomodifying Effect of Fucoidan from the Brown Alga S. cichorioides (ScF) with Pacificusoside D from the Starfish S. pacificus (SpD) on 3D Invasion of SK-MEL-2 Spheroids

The effectiveness of three-component combination of fucoidan ScF from *S. cichorioides* with X-ray radiation, and pacificusoside D (SpD) from *S. pacificus* (FXS combination) at low non-toxic concentrations on 3D invasion of SK-MEL-2 spheroids was evaluated using the 3D tumor spheroid invasion assay. Three days post-initiation SK-MEL-2 spheroids, treated with FXS combination, were embedded into gelatin matrix, which provides a semi-solid structure enriched with actin filaments and specific adhesion proteins. The invasion area was measured after 72 h of spheroids incubation. Three-dimensional SK-MEL-2 cells were shown to characterized high invasive potential (the area of spheroids increased by more than 10 times in 72 h in gelatin matrix, compared with the initial control of 0 h) (Figure 6a). It was found that ScF (50, 100, and 200 μg/mL) sensitized the SK-MEL-2 spheroids to X-ray (2 Gy) irradiation and pacificusoside D SpD (1 μM) causing significant reduction of their invasive potential of by 59%, 69%, and 82%, respectively, after 72 h of spheroids treatment (Figure 6a,b). Synergism of combined radiomodifying action of fucoidan and pacificusoside D (CI = 0.39; 0.26; 0.14) on the 3D invasion of human melanoma cells was observed (Figure 6c).

### 3.6. The Molecular Mechanism of the Combined Radiomodifying Effect of Fucoidan from the Brown Alga S. cichorioides (ScF) with Pacificusoside D from the Starfish S. pacificus (SpD)

The metastatic foci are characterized by a combination of molecular changes which are accumulated as result of deregulation a number of genes and products of their expression during the cancer progression [32,33]. The most studied molecular factors of cancer metastasis include proteins responsible for the apoptosis [34]. In this regard, we checked the idea that the combined treatment of SK-MEL-2 spheroids by fucoidan ScF from *S. cichorioides* in combination with X-ray and pacificusoside D SpD from *S. pacificus* (FXS combination) would induce apoptosis of melanoma cells.

The individual treatment of SK-MEL-2 spheroids by ScF (200 µg/mL), SpD (1 µM), and X-ray irradiation (2 Gy) slightly influenced the caspase 9, 7, and 3 expression levels, but their cleaved forms did not appear (Figure 7a). The FXS combination was found to induce apoptosis of melanoma cells by activating initiator caspase 9, effector caspases 7 and 3, and cleaved activated form caspase 3 (Figure 7a).

Under the individual treatment of SK-MEL-2 spheroids by ScF (200 µg/mL), SpD (1 µM), and X-ray irradiation (2 Gy) a slight degradation of DNA, giving the observed objects the appearance of comets was detected (Figure 7b, indicated by white arrow) FXS combination led to an increasing number of DNA reputes and, consequently, DNA comets (Figure 7b). These findings confirmed that fucoidan from the brown alga *S. cichorioides* in combination with triterpene glycoside at low non-toxic concentrations were able to enhance the apoptosis induced by X-ray irradiation by the activation of caspases which cleave, and thereby inactivate PARP, preventing the repair of DNA damage in melanoma cells.

## 4. Discussion

Cancer is a frightening disease and a serious medical problem that does not discern age, sex, race, ethnicity, or socio-economic status [35,36]. Melanoma is known to be a very aggressive type of cancer which is realized through several signaling pathways regulating cells growth, survival, and metastasis [2]. Given that metastatic melanoma is resistant to chemo- and radiotherapy and is associated with a poor patient prognosis, it is imperative to continue research into its optimal therapeutic methods [1]. The promising approach to increase the effectiveness of treatment modalities of malignant melanoma is the using of the nutraceutical compounds, as well as marine natural compounds, which are able sensitizing the cancer cells to radiation (radiomodifying effect) without affecting normal cells.

It was reported that dietary compounds or nutraceuticals such as diferuloylmethane (curcumin), plant hytoalexin (resveratrol), pentacyclic triterpenoid isolated from the root of *Tripterygium wilfordii* (celastrol) radiosensitized different types of cancer cells through the modulation of cell cycle, induction of apoptosis, and the regulation of TNF-α/NFκΒ and MAPK kinases activation [37]. Since the sulfated polysaccharides of brown algae, fucoidans, their derivatives, and glycosides from the starfishes, exhibited potent cytotoxic, anti-proliferative, anti-migratory, and anti-metastatic activities against human cancer cells [12,19,38,39] we check the idea that fucoidan from the brown alga *S. cichorioides* in combination with triterpene glycoside, pacificusoside D, from the starfish *S. pacificus* would be able to increase the inhibiting effect of low non-toxic doses X-ray radiation resulting in effective suppression of the viability and invasion of 3D melanoma cells SK-MEKL-2.

In the present study, we used the 2,4-O-sulfated 1,3-α-L-fucan with a small proportion of 1,4-linked α-L-fucopyranose residues and a small amount of single α-L-fucose residues in the branches (ScF), the product of autohydrolysis of fucoidan (ScF_AH) which was a 4-sulfated α-(1→3)-L-fucan, and triterpene glycoside, pacificusoside D (SpD), isolated from the starfish *S. pacificus*. It was demonstrated that ScF significantly decreased the proliferation rate of 3D melanoma cells SK-MEL-2 without affecting normal epidermal cells JB6 Cl41 at concentrations up to 800 µg/mL. On the other hand, its derivative ScF_AH was found to slightly influence the viability of tested cell lines. Probably, selective desulfation at position 2 and partial depolymerization of native fucoidan ScF during autohydrolysis led to decreasing of activity of derivative. SpD possessed significant cytotoxic effect against 3D SK-MEL-2 cells with high index of selectivity (SI = 4). It has been shown that fucoidans from different species of brown algae possessed an inhibiting action on the proliferation of various types of human cancer cell lines: leukemia, melanoma, colorectal carcinoma, hepatoma, lung cancer, and breast adenocarcinoma [8]. Fucoidans were reported to mediate their anti-proliferative effect at wide range of concentrations which is mainly depended on the structural characteristics of these polysaccharides [10]. The investigations of the cytotoxic activity of triterpene glycosides from starfishes are very limited. Recently, we determined that the pacificusosides D–K and cucumarioside D exhibited strong cytotoxicity against human melanoma cell SK-MEL-2, SK-MEL-28, and RPMI-7951 in 2D cell culture. The IC_50_ of pacificusoside D against 2D SK-MEL-2, SK-MEL-28, and RPMI-7951 was 0.7 µM, 8.4 µM, and 26.7 µM, respectively [22].

Based on the individual effect of ScF, ScF_AH, SpD, and X-ray, different variations of two-components and three-components treatment were tested to select the most effective regime for inhibition of viability and invasion of melanoma cells. The data obtained revealed that the pre-treatment of 3D SK-MEL-2 cells by fucoidan ScF was found to greatly sensitize cells to X-ray irradiation and enhance the inhibitory effect of triterpene glycoside SpD resulting in significant suppression of proliferation and invasion of human melanoma cells. Nowadays, the investigations devoted to the combination of fucoidans from the brown algae with chemotherapeutic drugs to destroy cancer are developed rapidly. It has been shown that the low molecular weight fucoidan from the brown algae *Undaria pinnatifida* (Mw < 10 kDa) exhibited synergistic effect with the DNA aptamer GroA and enhanced its antiproliferative and proapoptotic activities [40].

Another group of researchers determined the combined action of fucoidan from the brown algae *U. pinnatifida* (UPF) or *Fucus vesiculosus* (FVF) (commercial products of the Sigma-Aldrich company) and tamoxifen or paclitaxel against breast and prostate cancer in vivo [41]. UPF or FVF in combination with tamoxifen were found to have a synergistic effect in mice with breast cancer cells MCF-7 and ZR-75D. The FVF/tamoxifen drug combination was effective against the breast and prostate cancer cells. Additionally, the commercial fucoidan from *Fucus vesiculosus* (Sigma-Aldrich Company) in combination with chemotherapeutic drugs—cisplatin, doxorubicin, and taxol—significantly enhanced the cytotoxicity of these drugs against breast cancer cells MCF-7. Molecular mechanism of this action was associated with the regulation of the cell cycle and the induction of apoptosis. Moreover, fucoidan weakened toxicity of doxorubicin and cisplatin to normal breast cells. This effect not was found when the studied fucoidan was combined with taxol [42].

The fucoidan from the brown alga *Cladosiphon navaecaledoniae* Kylin was reported to enhance the antiproliferative and pro-apoptotic activities of cisplatin, tamoxifen or paclitaxel in breast cancer model MDA-MB-231 and MCF-7 via the regulation of the expression of anti-apoptotic proteins Bcl-xL and Mcl-1 and inhibition of the phosphorylation of protein kinases ERK1/2 and Akt in MDA-MB-231 cells but increased the ERK1/2 phosphorylation in MCF-7 cells [43].

The information on the radiomodifying effect of algal fucoidans and triterpene glycosides is very limited. It has been reported that fucoidan (40 μg/mL) from the brown alga *F. evanescens* and its derivative were able to selectively increase the sensitivity of human melanoma cells to radiation (4 Gy), which led to effective inhibition of proliferation and colony formation of radioresistant cancer cells. The polysaccharides were found to induce apoptosis of cancer cells by the activation of the mitochondrial apoptotic signaling cascade [30]. The ability of fucoidans to increase the sensitivity of different human cancer cell lines to radiation was supported by studies using heterogeneous fucoidan from *Dictyota dichotoma*, fucans from *L. longipes* and *S. cichorioides*, and galactofucans from *S. duplicatum*, *S. oligocystum*, and *S. feldmannii* [44]. The polysaccharides increased the effect of irradiation on cancer cells and safeguard the survival of normal cells.

It was reported that ginsenoside 20(S)-Rg3 from herbal plant ginseng in combination with curcumin enhanced the inhibiting effect of radiation on cell viability in breast cancer cells MDA-MB-231. The molecular mechanism of radiosensitizing action was related to the apoptosis induction [45].

The progression of melanoma is associated with uncontrolled cell proliferation, migration, and, consequently, the development of metastasis [32,33]. The regulation of cell death is critical for cancer cells to survive during metastasis [34]. That is why, in this work, the molecular mechanism of combined action of ScF, X-ray, and SpD was investigated in the focus of apoptosis induction.

Apoptosis is a type of programmed cell death that is characterized by cell membrane blebbing, cell shrinkage, nuclear fragmentation, chromatin condensation, and chromosomal DNA fragmentation [46]. There are two basic apoptotic signaling pathways: the extrinsic and the intrinsic pathways. In the intrinsic pathway, the functional consequence of pro-apoptotic signaling is mitochondrial membrane perturbation and release of cytochrome c in the cytoplasm, where it forms a complex or apoptosome with apoptotic protease activating factor 1 (APAF1) and the inactive form of caspase 9. This complex hydrolyzes adenosine triphosphate to cleave and activate caspase 9. The initiator caspase 9 then cleaves and activates the executioner caspases 3, 6, and 7, resulting in cell apoptosis. The findings of our study confirmed that fucoidan from the brown alga *S. cichorioides* in combination with pacificusoside D from the starfish *S. pacificus* at low non-toxic concentrations were able to enhance the apoptosis induced by X-ray irradiation by the activation of initiator caspase 9 and executor caspases 7 and 3, which cleave and thereby inactivate PARP, preventing the repair of DNA damage of 3D melanoma cells SK-MEL-2. An animal study is necessary to optimize the combined treatment schedule by fucoidan with pacificusoside D and X-ray and evaluate their pharmacodynamic impact in vivo.

## 5. Conclusions

In conclusion, more than twenty different combinations of fucoidan, pacificusoside D, and X-rays were tested in the model of viability and invasion of 3D SK-MEL-2 to determine the most effective therapeutic scheme for 3D melanoma cells SK-MEL-2. The most effective treatment scheme for SK-MEL-2 spheroids was found to be the pre-treatment of spheroids with fucoidan followed by irradiation by X-ray and the consequent treatment of cells by triterpene glycoside. The molecular mechanism of the radiomodifying effect of fucoidan in combination with pacificusoside D was associated with the induction of intrinsic apoptosis of melanoma cells. We believe that the combination of the unique biological properties of brown algae and starfish metabolites in radiochemotherapy might contribute to the development of a highly effective therapeutic approach for malignant melanoma. However, the optimization of the combined treatment conditions for further studies in vivo is needed.

## Figures and Tables

**Figure 1 biomolecules-13-00419-f001:**
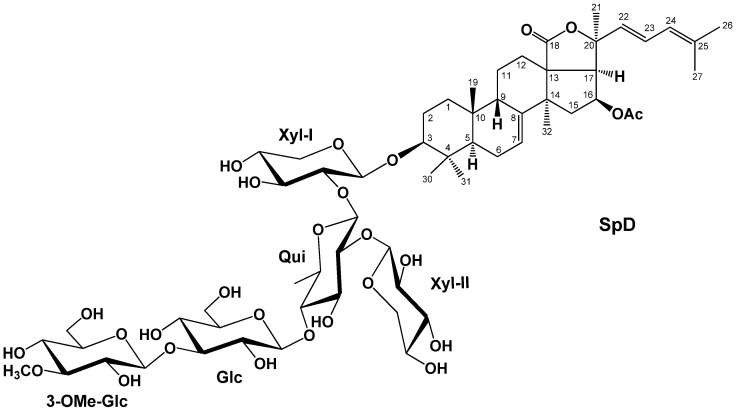
The structure of pacificusoside D (SpD) from the starfish *S. pacificus*.

**Figure 2 biomolecules-13-00419-f002:**
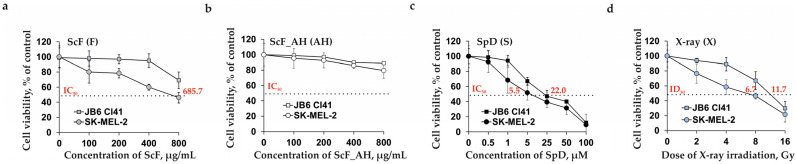
The effect of fucoidan from the brown alga *S. cichorioides* (ScF), its derivative ScF_AH (AH), pacificusoside D from starfish *S. pacificus* (SpD), and X-ray irradiation on viability of 3D mouse epidermal cells JB6 Cl41 and human melanoma cells SK-MEL-2. JB6 Cl41 and SK-MEL-2 spheroids were treated by (**a**) ScF (F) (100, 200, 400, and 800 µg/mL); (**b**) ScF_AH (AH) (100, 200, 400, and 800 µg/mL); (**c**) SpD (S) (0.5, 1, 5, 25, 50, and 100 µM); (**d**) X-ray (2, 4, 8, and 16 Gy) and incubated for 24 h. The cell viability was estimated using the MTS assay. Data are represented as the means ± SD as determined from triplicate experiments.

**Figure 3 biomolecules-13-00419-f003:**
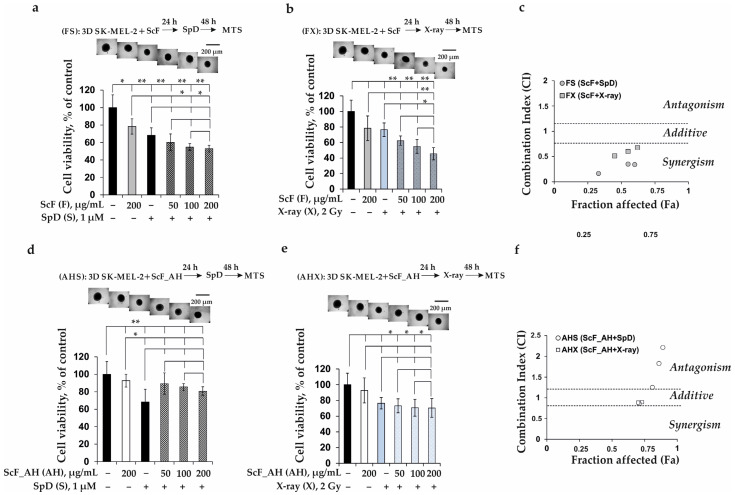
Two-component combined effect of fucoidan from *S. cichorioides* (ScF) with pacificusoside D from *S. pacificus* (SpD) or X-ray and the product of autohydrolysis of fucoidan ScF_AH (AH) with SpD (D) or X-ray on the viability of 3D human melanoma cells SK-MEL-2. SK-MEL-2 spheroids were treated by (**a**) ScF (50, 100, and 200 µg/mL) in combination with SpD (1 µM) (FS combination); (**b**) ScF (50, 100, and 200 µg/mL) in combination with X-ray (FX combination); (**d**) ScF_AH (50, 100, and 200 µg/mL) in combination with SpD (1 µM) (AHS combination); (**e**) ScF_AH (50, 100, and 200 µg/mL) in combination with X-ray (AHX combination) for total 72 h. The cell viability was estimated using MTS assay. Photographs (*n* = 6, where *n* = number of photographs) of each spheroid were taken using the ZOE™ Fluorescent Cell Imager. Data show the mean of three independent experiments ± SD. A one-way ANOVA and Tukey’s HSD test for multiple comparisons indicated the statistical significance (* *p* < 0.05 and ** *p* < 0.01). (**c**,**f**) Type of combined action of FS, FX, AHS, and AHX combinations calculated using Compusyn 1.0.1 software (ComboSyn, Inc., USA). The combined index (CI) is a quantitative measure of the degree of interaction between test compounds. CI equal to 0.9–1.1 is considered additive; a CI value greater than 1.1 represents antagonism; and CI values less than 0.7 indicate synergism.

**Figure 4 biomolecules-13-00419-f004:**
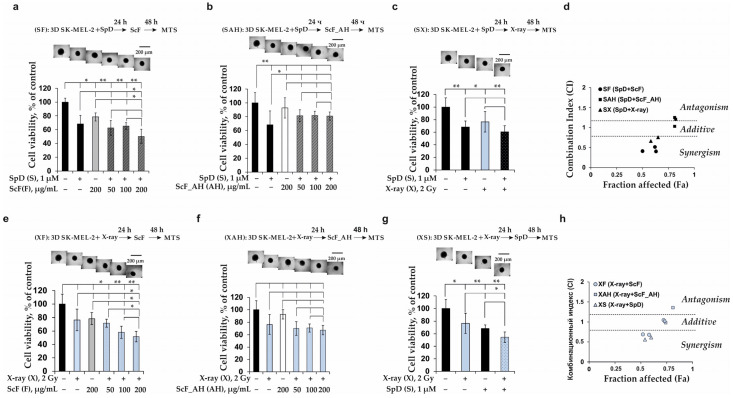
Two-component combined effect of pacificusoside D from *S. pacificus* (SpD) with fucoidan from *S. cichorioides* (ScF) or the product of autohydrolysis (ScF_AH) or with an X-ray, as well as an X-ray with SpD or with ScF or ScF_AH, on the viability of 3D human melanoma cells SK-MEL-2. SK-MEL-2 spheroids were treated by (**a**) SpD (1 µM) in combination with ScF (50, 100, and 200 µg/mL) (SF combination); (**b**) SpD (1 µM) in combination with ScF_AH (50, 100, and 200 µg/mL) (SAH combination); (**c**) SpD (1 µM) in combination with X-ray (2 Gy) (SX combination); (**e**) X-ray (2 Gy) in combination with ScF (50, 100, and 200 µg/mL) (XF combination); (**f**) X-ray (2 Gy) in combination with ScF_AH (50, 100, and 200 µg/mL) (XAH combination); (**g**) X-ray (2 Gy) in combination with SpD (1 µM) (XS combination) for total 72 h. The cell viability was estimated using the MTS assay. The cell viability was estimated using the MTS assay. Photographs (*n* = 6, where *n* = number of photographs) of each spheroid were taken using the ZOE™ Fluorescent Cell Imager. Data show the mean of three independent experiments ± SD. A one-way ANOVA and Tukey’s HSD test for multiple comparisons indicated the statistical significance (* *p* < 0.05 and ** *p* < 0.01). (**d**,**h**) Type of combined action of SF, SAH, SX, XF, XAH, and XS combinations calculated using Compusyn 1.0.1 software (ComboSyn, Inc., USA).

**Figure 5 biomolecules-13-00419-f005:**
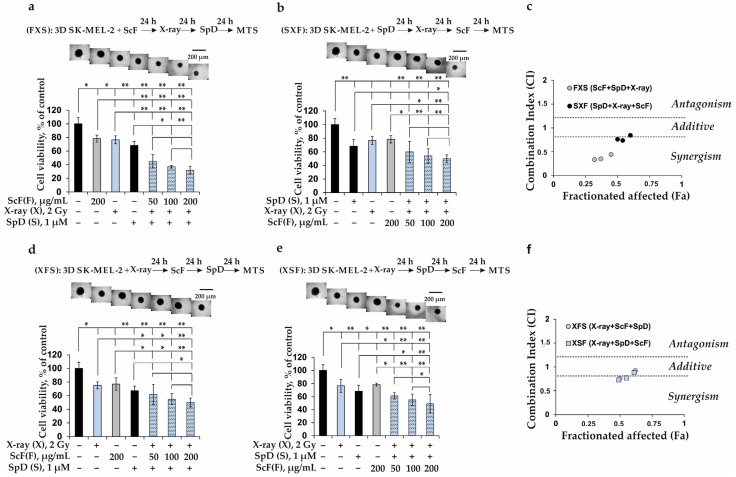
Three-component combined effect of fucoidan from *S. cichorioides* (ScF) with pacificusoside D from *S. pacificus* (SpD) and X-ray on the viability of 3D human melanoma cells SK-MEL-2. SK-MEL-2 spheroids were treated by (**a**) ScF (50, 100, and 200 µg/mL) in combination with X-ray (2 Gy) and SpD (1 µM) (FXS combination); (**b**) SpD (1 µM) in combination with X-ray (2 Gy) and ScF (50, 100, and 200 µg/mL) (SXF combination); (**d**) X-ray (2 Gy) in combination with ScF (50, 100, and 200 µg/mL) and SpD (1 µM) (XFS combination) (**e**) X-ray (2 Gy) in combination with SpD (1 µM) and ScF (50, 100, and 200 µg/mL) (XSF combination) for total 72 h. The cell viability was estimated using the MTS assay. The cell viability was estimated using the MTS assay. Photographs (*n* = 6, where *n* = number of photographs) of each spheroid were taken using the ZOE™ Fluorescent Cell Imager. Data show the mean of three independent experiments ± SD. A one-way ANOVA and Tukey’s HSD test for multiple comparisons indicated the statistical significance (* *p* < 0.05 and ** *p* < 0.01). (**c**,**f**) Type of combined action of FXS, SXF, XFS, and XSF combinations calculated using Compusyn 1.0.1 software (ComboSyn, Inc., USA).

**Figure 6 biomolecules-13-00419-f006:**
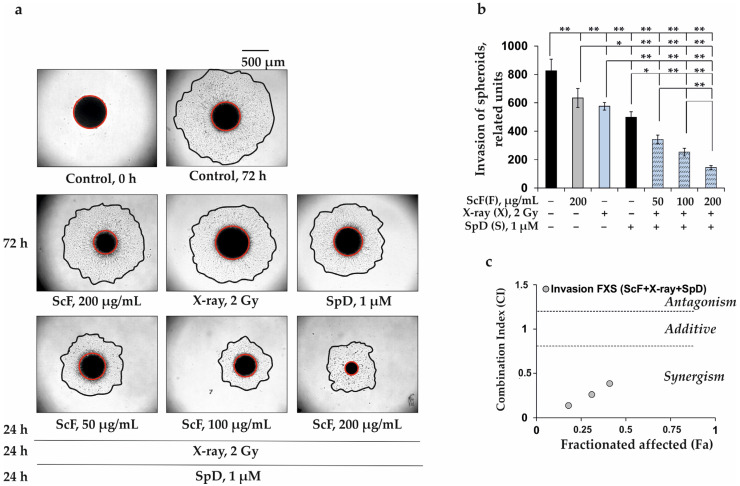
The combined radiomodifying effect of fucoidan from *S. cichorioides* (ScF) with X-ray and pacificusoside D from *S. pacificus* (SpD) on the invasion of 3D human melanoma cells SK-MEL-2. (**a**) Photographs of SK-MEL-2 spheroids (40×, scale 500 µm) after invasion into the gelatin matrix for 72 h treated with ScF (50, 100, and 200 µg/mL) in combination with X-ray (2 Gy) and SpD (1 μM). (**b**) Quantification of invasion of spheroids treated with the combination of investigated compounds was performed using the ImageJ program and was determined as the difference between the area of invasion of spheroid cells and the area of the spheroid. Data show the mean of three independent experiments ± SD. A one-way ANOVA and Tukey’s HSD test for multiple comparisons indicated the statistical significance (* *p* < 0.05 and ** *p* < 0.01). (**c**) The Combination Index (CI) for the interaction between ScF, X-ray, and SpD was calculated using the Compusyn software (ComboSyn, Inc.).

**Figure 7 biomolecules-13-00419-f007:**
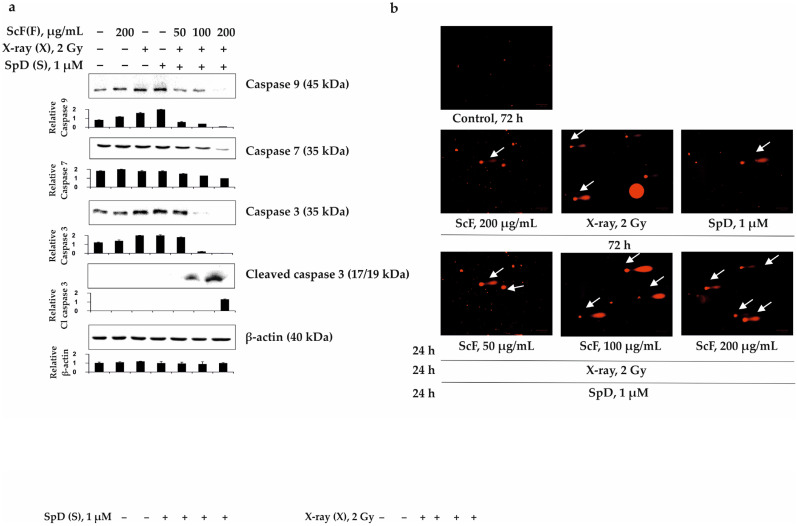
The combined radiomodifying effect of fucoidan from *S. cichorioides* (ScF) with X-ray and pacificusoside D from *S. pacificus* (SpD) on the induction of apoptosis and DNA degradation of 3D human melanoma SK-MEL-2. (**a**) Western blot analyses of caspase 9, 7, and 3 activation, and cleaved caspase 3 expression in spheroids treated by ScF in combination with X-ray and SpD for 72 h. β-actin was used as an internal control. Relative band density was measured using Image Lab™ Software 4.1. (**b**) DNA degradation of 3D SK-MEL-2 cells treated ScF in combination with X-ray and SpD for 72 h determined by the DNA comet method. Photographs (*n* = 3, where *n* = number of photographs, scale 100 µm) of each spheroid were taken using the ZOE™ Fluorescent Cell Imager.

## Data Availability

The data presented in this study are available on request from the corresponding author.

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
