# Peer review of "Combined Radiomodifying Effect of Fucoidan from the Brown Alga Saccharina cichorioides and Pacificusoside D from the Starfish Solaster pacificus in the Model of 3D Melanoma Cells"

_biomolecules, 2023, doi:10.3390/biom13030419_

Round 1

Reviewer 1 Report

An interesting, well designed study and well written research paper in which a novel combination of marine bioactives and radiation have been assessed in vitro, with a 3D melanoma model.

See Line 415  sp. ''radiomodifying effect.

Discussion

Refer to the next step- optimisation of in-vivo treatment schedule in this part, as well as the conclusion.

Whilst the translation of research to actual therapy is always challenging, do the authors see potential for oral delivery as an adjunct to standard cancer therapy ?  

The dietary or 'nutraceutical' approach to sensitization in cancer therapy has been discussed by several other groups, and is perhaps interesting in the context of natural product development discussed here.  

In the discussion it may be interesting to consider as a comparison to the starfish triterpene, the triterpene celastrol from Tripterygium wilfordii

Calvaruso M, Pucci G, Musso R, Bravatà V, Cammarata FP, Russo G, Forte GI, Minafra L. Nutraceutical Compounds as Sensitizers for Cancer Treatment in Radiation Therapy. Int J Mol Sci. 2019 Oct 23;20(21):5267. doi: 10.3390/ijms20215267

and perhaps the triterpenes from ginseng, which have also been shown to act as radiosensitizers.eg Changizi V, Co-treatment with Ginsenoside 20(S)-Rg3 and Curcumin increases Radiosensitivity of MDA-MB-231 Cancer Cell Line. Iran J Med Sci. 2021 Jul;46(4):291-297. doi: 10.30476/ijms.2020.83977.1334.

Author Response

Dear Editor and Reviewers,

Thank you for careful review of our manuscript. We are very grateful for your censorious remarks and useful comments. We have revised our results and added the data in accordance with your comments. Enclosed please find our latest version of manuscript that is the revised version of our paper.

Comment 1. See Line 415  sp. ''radiomodifying effect.

Answer 1. We agree with Reviewer’s comment. It was corrected as suggested by Reviewer.

Comment 2. Discussion. Refer to the next step- optimisation of in-vivo treatment schedule in this part, as well as the conclusion.

Answer 2. The sentence “An animal study is necessary to optimize the combined treatment schedule by fucoidan with pacificusoside D and X-ray and evaluate their pharmacodynamic impact in vivo.” was added to the Discussion. There was the sentence “However the optimization of the combined treatment conditions for further studies in vivo is needed.” in the Conclusion.

Comment 3. Whilst the translation of research to actual therapy is always challenging, do the authors see potential for oral delivery as an adjunct to standard cancer therapy ?

Answer 3. We are planning to use oral delivery of fucoidan and intraperitoneal injection of triterpene glycoside from starfish, since both of these types of administration were approved for fucoidan from brown alga Fucus evanescens [data not published] and holotoxin A1 from sea cucumber Apostichopus japonicus [Malyarenko OS, Ivanushko LA, Chaikina EL, et al. In Vitro and In Vivo Effects of Holotoxin A1 From the Sea Cucumber Apostichopus japonicus During Ionizing Radiation. Natural Product Communications. 2020;15(6). doi:10.1177/1934578X20932033] earlier.

Comment 4. The dietary or 'nutraceutical' approach to sensitization in cancer therapy has been discussed by several other groups, and is perhaps interesting in the context of natural product development discussed here. In the discussion it may be interesting to consider as a comparison to the starfish triterpene, the triterpene celastrol from Tripterygium wilfordii.

Calvaruso M, Pucci G, Musso R, Bravatà V, Cammarata FP, Russo G, Forte GI, Minafra L. Nutraceutical Compounds as Sensitizers for Cancer Treatment in Radiation Therapy. Int J Mol Sci. 2019 Oct 23;20(21):5267. doi: 10.3390/ijms20215267

and perhaps the triterpenes from ginseng, which have also been shown to act as radiosensitizers.eg Changizi V, Co-treatment with Ginsenoside 20(S)-Rg3 and Curcumin increases Radiosensitivity of MDA-MB-231 Cancer Cell Line. Iran J Med Sci. 2021 Jul;46(4):291-297. doi: 10.30476/ijms.2020.83977.1334.

Answer 4. We agree with Reviewer’s comment. It was corrected as suggested by Reviewer. The sentence and “It was reported that dietary compounds or nutraceuticals such as diferuloylmethane (curcumin), plant hytoalexin (resveratrol), pentacyclic triterpenoid isolated from the root of Tripterygium wilfordii (celastrol) radiosensitized different types of cancer cells through the modulation of cell cycle, induction of apoptosis, and the regulation of TNF-α/NFκΒ and MAPK kinases activation [37].” and “It was reported that ginsenoside 20(S)-Rg3 from herbal plant ginseng in combination with curcumin enhanced inhibiting effect of radiation on cell viability in breast cancer cells MDA-MB-231. The molecular mechanism of radiosensitizing action was related to the apoptosis induction [45].”was added to the Discussion.

Reviewer 2 Report

The authors presented the findings of a study on the radiomodifying effect of fucoidan from the brown algae S. cichorioides and its autohydrolysis derivative in combination with the triterpene glycoside, pacificusoside D, from the starfish S. pacificus on the viability and invasion model of three-dimensional (3D) human melanoma cells SK-MEL-2. The study revealed the molecular mechanism of the researched biomolecules' combined radiomodifying impact, which is connected with the induction of apoptosis and DNA fragmentation in SK-MEL-2 cells. The article seems to be an interesting one and can be accepted after minor revision.

1. The article abstract does not completely present the manuscript's theme and hence fails to entice the reader. Modify it accordingly.

2. In the introduction, explain why this model was chosen and how your findings served to validate pre-existing knowledge in cancer therapies.

3. Proofread the work carefully to ensure proper English usage.

4. Update some of the references to more recent works.

Author Response

Dear Editor and Reviewers,

Thank you for careful review of our manuscript. We are very grateful for your censorious remarks and useful comments. We have revised our results and added the data in accordance with your comments. Enclosed please find our latest version of manuscript that is the revised version of our paper.

Comment 1. The article abstract does not completely present the manuscript's theme and hence fails to entice the reader. Modify it accordingly.

Answer 1. The abstract was corrected as suggested by Reviewer.

Comment 2. In the introduction, explain why this model was chosen and how your findings served to validate pre-existing knowledge in cancer therapies.

Answer 2. The paragraph about 3D culture model “In the past, the in vitro cytotoxic activity of synthetic and natural compounds was studied using established cancer cell lines grown as two-dimensional (2D) cultures characterized by a rapid, uncontrolled growth phenotype. However 2D cell cultures are not capable of mimicking the complexity and heterogeneity of clinical tumors as in vivo tumors grow in a three-dimensional (3D) conformation with a specific organization and architecture that a 2D monolayer cell culture cannot reproduce. Three-dimensional (3D) growth of immortalized established cell lines or primary cell cultures possess several in vivo features of tumors such as cell-cell interaction, hypoxia, drug penetration, response and resistance, and production/deposition of extracellular matrix [23].” was added to the Introduction Section.

Comment 3. Proofread the work carefully to ensure proper English usage.

Answer 3. The manuscript was carefully proofreaded.

Comment 4. Update some of the references to more recent works.

Answer 4. Some references were updated to more recent works. However references attributed to the methods used and structure elucidation of investigated compounds cannot be changed.